# Aldehyde-alcohol dehydrogenase forms a high-order spirosome architecture critical for its activity

Gijeong Kim [1], Liyana Azmi[2], Seongmin Jang[1], Taeyang Jung[1,3,4], Hans Hebert [3,4], Andrew J. Roe [2], Olwyn Byron [5] & Ji-Joon Song [1]*

Aldehyde-alcohol dehydrogenase (AdhE) is a key enzyme in bacterial fermentation, converting acetyl-CoA to ethanol, via two consecutive catalytic reactions. Here, we present a 3.5 Å resolution cryo-EM structure of full-length AdhE revealing a high-order spirosome architecture. The structure shows that the aldehyde dehydrogenase (ALDH) and alcohol dehydrogenase (ADH) active sites reside at the outer surface and the inner surface of the spirosome respectively, thus topologically separating these two activities. Furthermore, mutations disrupting the helical structure abrogate enzymatic activity, implying that formation of the spirosome structure is critical for AdhE activity. In addition, we show that this spirosome structure undergoes conformational change in the presence of cofactors. This work presents the atomic resolution structure of AdhE and suggests that the high-order helical structure regulates its enzymatic activity.

[1] Department of Biological Sciences, Korea Advanced Institute of Science and Technology (KAIST), Daejeon 34141, Korea. [2] Institute of Infection, Immunity and Inflammation, University of Glasgow, Glasgow G12 8QQ Scotland, UK. [3] School of Engineering Sciences in Chemistry, Biotechnology and Health, KTH Royal Institute of Technology, Novum SE-141 57, Sweden. [4] Department of Biosciences and Nutrition, Karolinska Institutet, S-141 83 Huddinge, Sweden. [5] School of Life Sciences, University of Glasgow, Glasgow G12 8QQ Scotland, UK. *email: songj@kaist.ac.kr

Aldehyde-alcohol dehydrogenase (AdhE) is a bifunctional aldehyde-alcohol dehydrogenase highly conserved in bacteria as well as fungus, algae, and protozoan parasites[1–3]. Previous work on AdhE has focused on its role in anaerobic conditions, where this multi-functional enzyme is essential for the fermentation of glucose to sustain the glycolytic pathway. The conversion of acetyl-CoA to ethanol is carried out in a two-step reaction performed by the two catalytic domains of AdhE. The N-terminal aldehyde dehydrogenase (ALDH) domain converts acetyl-CoA to acetaldehyde, which is then converted to ethanol by the C-terminal alcohol dehydrogenase (ADH) domain. AdhE is also a bidirectional enzyme and thus can utilize ethanol as a substrate to generate NADH and other carbon intermediates[4,5]. AdhE is also reported to have a third enzymatic function as a pyruvate formate-lyase, an enzyme that catalyses the conversion of pyruvate and coenzyme A to formate and acetyl-CoA[4].

AdhE is highly conserved amongst anaerobic bacteria such as primary fermenters (enterobacteria, clostridia) and acetogenic bacteria such as *Acetobacterium woodii*[5,6]. Deletion of AdhE in pathogenic *Escherichia coli* O157:H7 reduces bacterial virulence and induces overexpression of non-functional flagella[7]. This phenotype makes AdhE an attractive anti-virulence drug target. Thus, the high-resolution structure of AdhE is essential as a template for structure-based drug design. The N-terminal ALDH is linked to the C-terminal ADH by a linker and the proximity of these domains to each other is likely to enable substrate channelling for an improved rate of ethanol production[8]. This particular reaction is of interest to both human health and biotechnology due to the role of AdhE in the regulation of alcohol metabolism[9]. In addition, ethanol production via AdhE catalysis is widely studied as a prospect for renewable energy production[10,11]. Deletion of *adhE* is correlated with at least 90% loss of ethanol yield[12]. Despite the importance of AdhE, little is known about the molecular details of its high-order structure and the implications of this structure with regards to its enzymatic activity.

Structurally, AdhE is interesting as this 96 kDa protein oligomerises to form long filaments which can be visualised by electron microscopy. This was first reported by Kessler et al.[4] who showed that the arrangement of these filaments was influenced by the presence of the cofactors (NAD$^+$ and Fe$^{2+}$). The physiological role of these filaments, called spirosomes[13], is a mystery. Therefore, we sought to investigate the spirosome structure of *E. coli* AdhE using cryo-EM and its implication for the activity of the enzyme. Here, we present an atomic resolution cryo-EM structure of AdhE forming a spirosome and show that this spirosome formation is critical for AdhE activity. This work implicates the regulation of enzymatic activities by high-order structure formation in AdhE, and provides a platform upon which to design inhibitors against AdhE for anti-virulence drug design and to engineer AdhE for the purpose of enhanced production of alcohol.

## Results

**Cryo-EM structure of AdhE in spirosome form.** To examine the molecular architecture of AdhE, we purified full-length AdhE and fractionated it further via Superdex 200 gel-filtration chromatography wherein a broad elution profile was observed indicating the existence of various oligomeric states of the AdhE monomer (96.1 kDa) (Fig. 1a). To characterize the nature of these species, we selected fractions of presumed different molecular weights and examined them by negative stain electron microscopy (EM) (Fig. 1b). Fraction 1 contained a longer spirosome structure of length 25–100 nm and fraction 2 contained a shorter spirosome.

Lastly, fraction 3 comprised relatively small particles, possibly tetrameric or dimeric AdhE. We analyzed the length distribution of AdhE in fraction 1 showing that the length varies from 15 to 120 nm without any dominant population of one specific length (Supplmentary Fig. 1). These data indicate that AdhE can be isolated in a wide range of oligomeric states. To determine the atomic structure of AdhE, we undertook cryo-EM imaging of AdhE. Cryo-grids were prepared by plunge-freezing AdhE from fractions 1 or 2, which are dominated by different AdhE oligomeric states. First, a total of 1255 micrographs of the sample from the fraction 2 were collected using a Titan Krios 300 keV microscope with a Falcon III direct detector in electron counting mode. 160,830 particles out of 251,604 particles picked were further processed using cisTEM [14] to generate a 3.5 Å resolution cryo-EM map (Fig. 1c, d, Supplementary Fig. 2, Supplementary Fig. 3 and Supplementary Table 1). 2D class averages show clear secondary structure features and indicate the existence of a helical structure (Fig. 1c). Most of the side chains were resolved in the cryo-EM map, and we built atomic models of six complete AdhE molecules and two ADH domains (Fig. 2a, b and Supplementary Fig. 3). These molecules are stacked upon each other to form a right-handed helix with a 70 Å helical pitch and 150 Å diameter (Fig. 2). The cryo-EM structure of AdhE shows that the ALDH and ADH domains in the AdhE monomer are separated by a linker (residues 441–448) (Fig. 3a). Together with a β-hairpin protruding from the ALDH domain, the linker makes a three-stranded β-sheet stabilizing connections between the two catalytic domains (Fig. 3a). The structure of the ALDH domain is similar to other known ALDHs and is composed of two lobes. Each lobe has a canonical Rossman fold[15] formed by a β-sheet surrounded by helices forming a NADH$^+$ binding cleft, as observed in other dehydrogenases[16]. The ADH domain also consists of two lobes with a Fe$^{2+}$ and NADH$^+$-binding pocket between them, which is similar to other ADH domains[8]. Two AdhE monomers form a dimer in a head-to-head arm-crossing fashion (Fig. 3b). The three-stranded β-sheet in the linker forms a continual β-sheet interaction with the β-sheet within the ALDH domain from the other molecule (Fig. 3b). Subsequently, two dimers (four AdhE molecules) form one helical pitch via the interaction of ADH domains in a tail-to-tail manner (Fig. 3c). With this configuration, six AdhE molecules and two ADH domains at the top and the bottom of the helical structure comprise about one-and-a-half helical turns in our cryo-EM structure. By repeating the helical unit, AdhEs form into a spirosome structure, which might lead to activation of its biochemical activity by clustering enzymes.

**The spirosome topologically separtes ALDH and ADH activities.** To further investigate the nature of the spirosome structure of AdhE, we collected cryo-EM micrographs from the sample containing longer spirosomes (fraction 1) with a Talos Artica 200 keV microscope using a Falcon III direct detector in an integration mode. The ends of the spirosome molecules were manually picked and subsequently picked with helical auto-picking using Relion[17]. A final 39,443 particle set was further processed according to the helical reconstruction process to generate 2D class-averages (Fig. 4a). 2D class-averages were selected and 3D classification was subsequently undertaken to generate an 11.2 Å cryo-EM structure of helical AdhE (Fig. 4b and Supplementary Fig. 4). Having the high-resolution cryo-EM structure of one-and-a-half helical turns, we were able to reconstitute a continual AdhE spirosome structure based on the cryo-EM structure resulting from helical reconstitution (Fig. 4b). In the spirosome structure, there are inter-helical interactions between ADH domains near the ADH catalytic site (Fig. 4b). Specifically, residues N492 and R488 from two ADH domains interact with

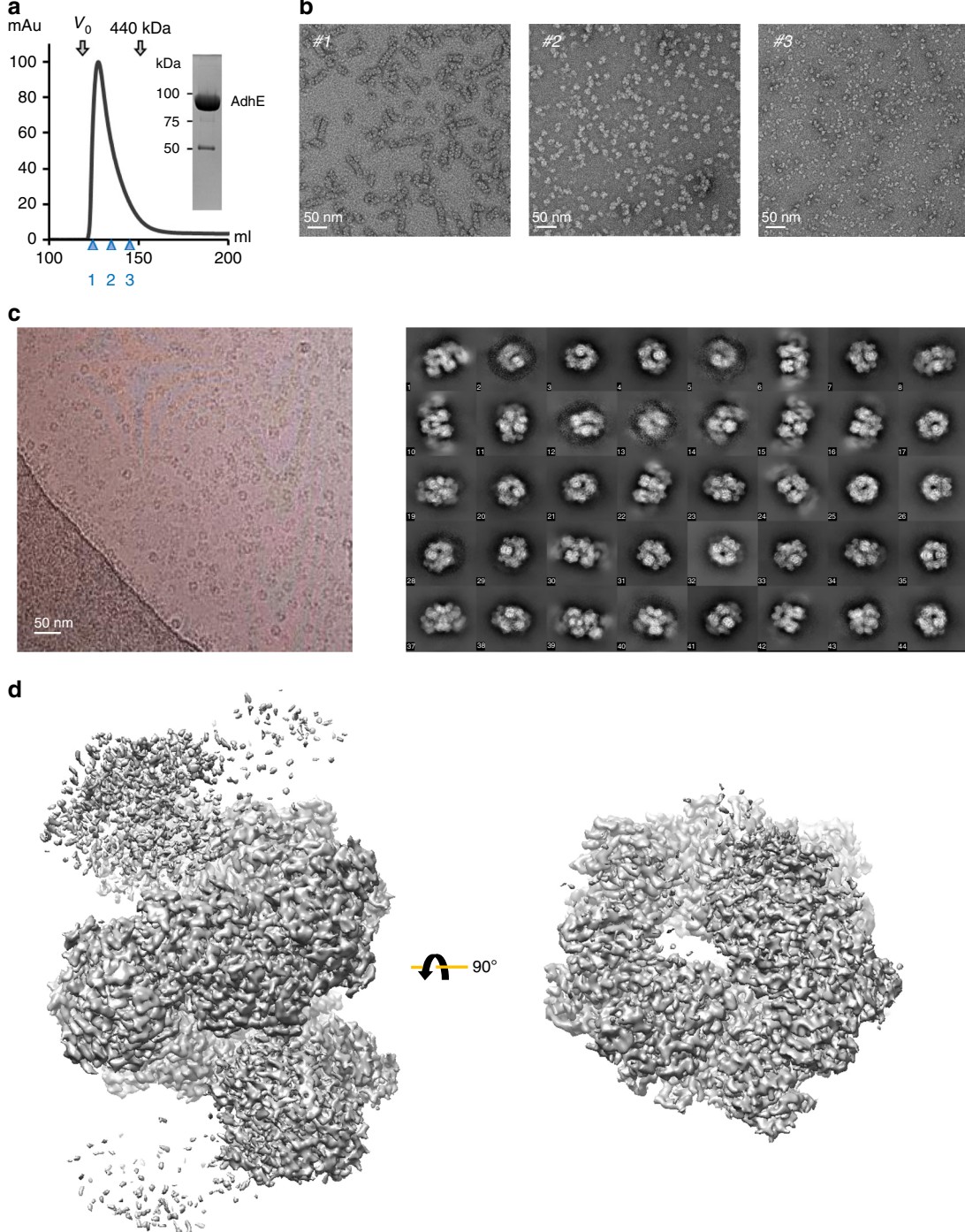

**Fig. 1** Cryo-EM analysis of AdhE. **a** AdhE eluted across a broad molecular weight range in Superdex 200 gel-filtration. The void volume ($V_0$) and elution volume for a molecular weight marker are indicated above the elution profile, and the fractions examined by negative stain EM are indicated below the profile. An SDS-PAGE gel shows the purity of AdhE used as input for the gel-filtration. **b** Negative stain EM analysis of fractions 1, 2 and 3 showing that AdhE forms a range of higher order structures. Scale bar 50 nm. **c** A representative micrograph (left) and 2D class averages (right). **d** Cryo-EM maps of AdhE in two different orientations. The residual density at the top and the bottom in the left panel indicates the helical property of AdhE

each other, and Q821 interacts with the backbone of the loop between residues 816–821 from the other ADH domain. Interestingly, the NADH binding pocket is located near to the site of interaction between two ADH domains, suggesting that AdhE spirosome formation might affect its activity. In the spirosome structure, ALDH domains, as well as ADH domains from adjacent subunits, are clustered, which might render the substrate easily accessible to each activity (Fig. 4c). To further investigate

the implication of the spirosome structure of AdhE, the plausible active sites of ALDH and ADH were highlighted on the helical structure. This practice reveals that ALDH active sites are located towards the outer surface of the helical structure while ADH active sites reside towards the inner surface. Thus, these two activities are topologically separated in the spirosome architecture (Fig. 4d). To further examine the properties of AdhE spirosome in solution, we undertook small angle X-ray scattering (SAXS).

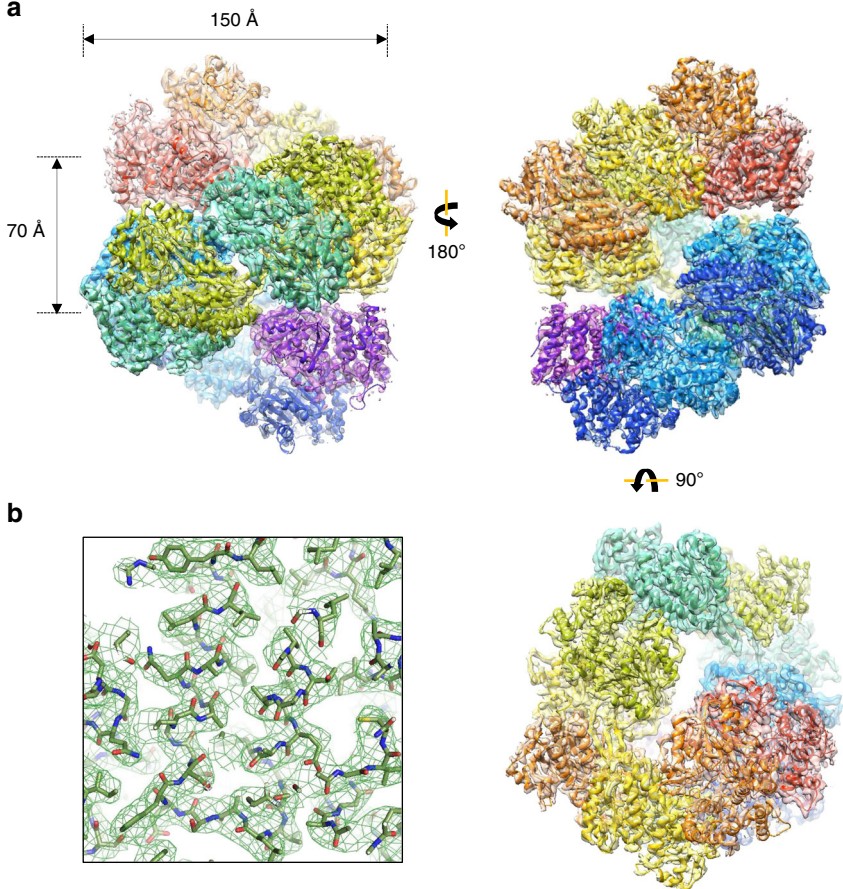

**Fig. 2** AdhE forms a spirosome structure. **a** The cryo-EM structure of AdhE with six AdhE and two ADH domains fitted to the cryo-EM map. Each AdhE subunit is in a different colour. The structure is viewed from different angles (left, right and below). **b** The cryo-EM map with the refined model

SAXS data for AdhE were obtained in batch mode without further fractionation by SEC. SAXS profiles were computed with the FoXS server[18,19] for a series of models constructed from the continual spirosome structure (two of which are shown in Fig. 4e). This analysis indicates that while the experimental data derive from a polydisperse AdhE sample, the overall shape of the SAXS curve is broadly consistent with that computed for a spirosome model and is better described by the profile computed for the 24-mer model (the largest tested) than for the 12-mer or any lesser oligomers. This analysis is also consistent with the average length of spirosomes (460 Å), which corresponds to a 24 mer comprising 7 pitches (1 pitch = 70 Å). Overall, these combined data show that AdhE forms into a spirosome structure by which the activity of AdhE might be activated.

**The spirosome undergoes structural changes with NADH.** AdhE spirosomes in bacterial lysate have previously been observed to occupy either an 'open' or a 'closed' conformation, depending on the presence or absence of cofactors ($Fe^{2+}$ and $NAD^+$)[20]. Work by Kessler and colleagues[20] showed via negative stain EM that, in the presence of 5 mM $NAD^+$ and 0.3 mM $FeSO_4$, AdhE spirosomes relax to a looser helical assembly, changing length from 40–120 nm to 60–220 nm, and diameter from $15 \pm 2$ nm to $13.5 \pm 1$ nm. Here, we confirm a change in spirosome conformation in the presence of cofactors using negative stain EM and SAXS (Fig. 5). Compared with the negative stain image of AdhE in the absence of the cofactors, which is well fitted by our cryo-EM structure (Supplementary Fig. 5), the spirosome structure of AdhE in the presence of $NAD^+$ and $FeSO_4$

seems to be extended along the long axis resulting in a narrower width and longer pitch than AdhE without cofactors (Fig. 5a), suggesting that spirosomes in an extended conformation are formed upon addition of the cofactors.

To observe the global conformational change in the entire population of spirosome species in each sample in the presence and absence of cofactors in solution, SAXS data for AdhE in its apo form and with different combinations of cofactors (NADH + $FeSO_4$, NADH, $NAD^+$ + $FeSO_4$ and $NAD^+$) were acquired in batch mode without fractionation by SEC. A dramatic shift in the reciprocal space position of the feature characteristic of AdhE spirosomes from $q = 0.086$ Å$^{-1}$ for apo- AdhE to $q = 0.075$ Å$^{-1}$ is observed upon the addition of all cofactor combinations (Fig. 5b and Supplementary Fig. 6). This feature is consistent with the helical pitch of the cryo-EM structure of spirosomes and translates to a relaxation in real-space from 73.0 to 83.7 Å upon cofactor addition. These data suggest that the flexibility of the spirosome structure might be implicated in its activity.

**The spirosome structure is required for AdhE activity.** Next, we asked whether spirosome structure has implications for AdhE activity. To design a mutant disrupting the helical formation, the interface between AdhE molecules was examined. F670 in the ADH domain is inserted into a hydrophobic pocket formed by F462, I460, and I712 of the other ADH domain and holds the ADH-ADH domains together (Fig. 6a). To disrupt the AdhE self-association, F670 was mutated to several amino acids: Val (V), Ala (A), and Glu (E) (Supplementary Fig. 7). All mutants eluted much later in gel-filtration indicating that oligomerisation was

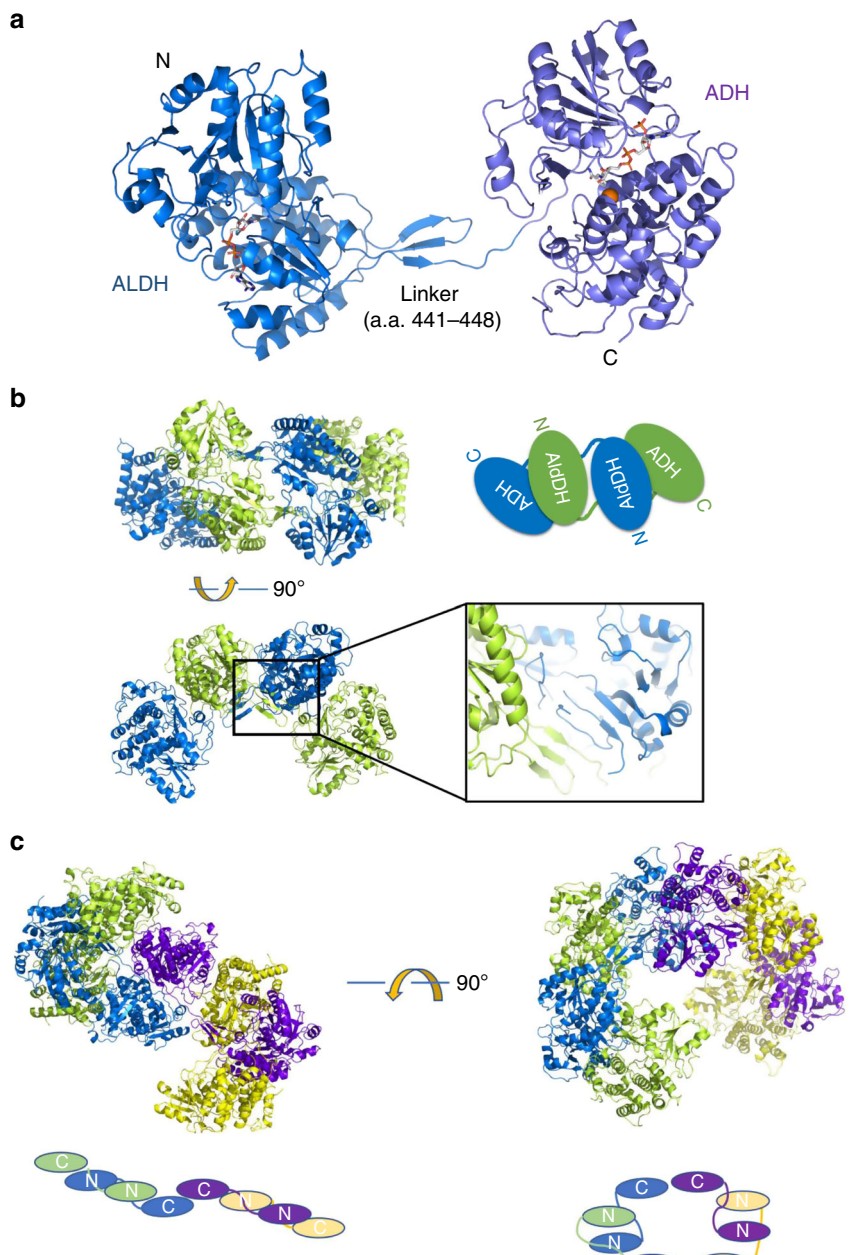

**Fig. 3** Hierarchical formation of AdhE spirosome from a monomer. **a** A full-length AdhE monomer is shown as a ribbon model. The N-terminal ALDH domain (royal blue) and C-terminal ADH domain (light purple) are linked by a short β-sheet composed of one β-strand from the linker and two β-strands from the ALDH domain. $NAD^+$ and $Fe^{2+}$ were modeled from other alcohol dehydrogenase structures (PDB IDs: 3MY7, 3ZDR). **b** AdhE forms a dimer by interacting in a head-to-head arm-crossing fashion. The short β-sheet in the linker forms a continual β-sheet with the β-sheet from the ALDH domain. **c** ADH domains from the AdhE dimer interact in a tail-to-tail manner and a total of four AdhE molecules make one helical turn

disrupted in all the mutants (Fig. 6b). The gel-filtration profiles of all mutants showed a symmetric single peak and the mutants behaved well during purification indicating that the mutations did not disrupt the global structure of AdhE (Fig. 6b and Supplementary Fig. 7). To examine if the spirosome structure was disrupted in these mutants, they were examined by negative stain EM, analytical ultracentrifugation (AUC) and SAXS. Compared with the wild-type (WT), the spirosome structure was not observed in negative stain EM for any of the mutants (Fig. 6c), suggesting that the hydrophobic interaction mediated by F670 is critical for spirosome formation. Analysis of sedimentation velocity (SV)-AUC data for $AdhE_{F670A}$ (F670A), $AdhE_{F670V}$ (F670V) and $AdhE_{F670E}$ (F670E) demonstrates that large species

remain in all three mutant samples but that, in comparison with AdhE from *Yersinia pestis* ($AdhE_{YP}$), the population becomes dominated by two lower s species (peaks '1' and '2' in Fig. 6d) at ≈ 5.1 S and 7.9 S. The smallest species in $AdhE_{YP}$ has $s_{20,w} = 7.6$ S (peak '3' in Fig. 6d). $s_{20,w}$ was calculated using SOMO[21] for the coordinates of monomeric AdhE extracted from the high-resolution cryo-EM structure giving 5.1 S, in perfect agreement with that observed for peak '1' in Fig. 6d, confirming that WT $AdhE_{YP.}$ is devoid of monomer.

The F670E mutant was fractionated by SEC and data for one fraction which was observed to be monodisperse by virtue of constant radius of gyration ($R_g$) (Supplementary Fig. 9a) were further analysed. Guinier analysis and pairwise distance

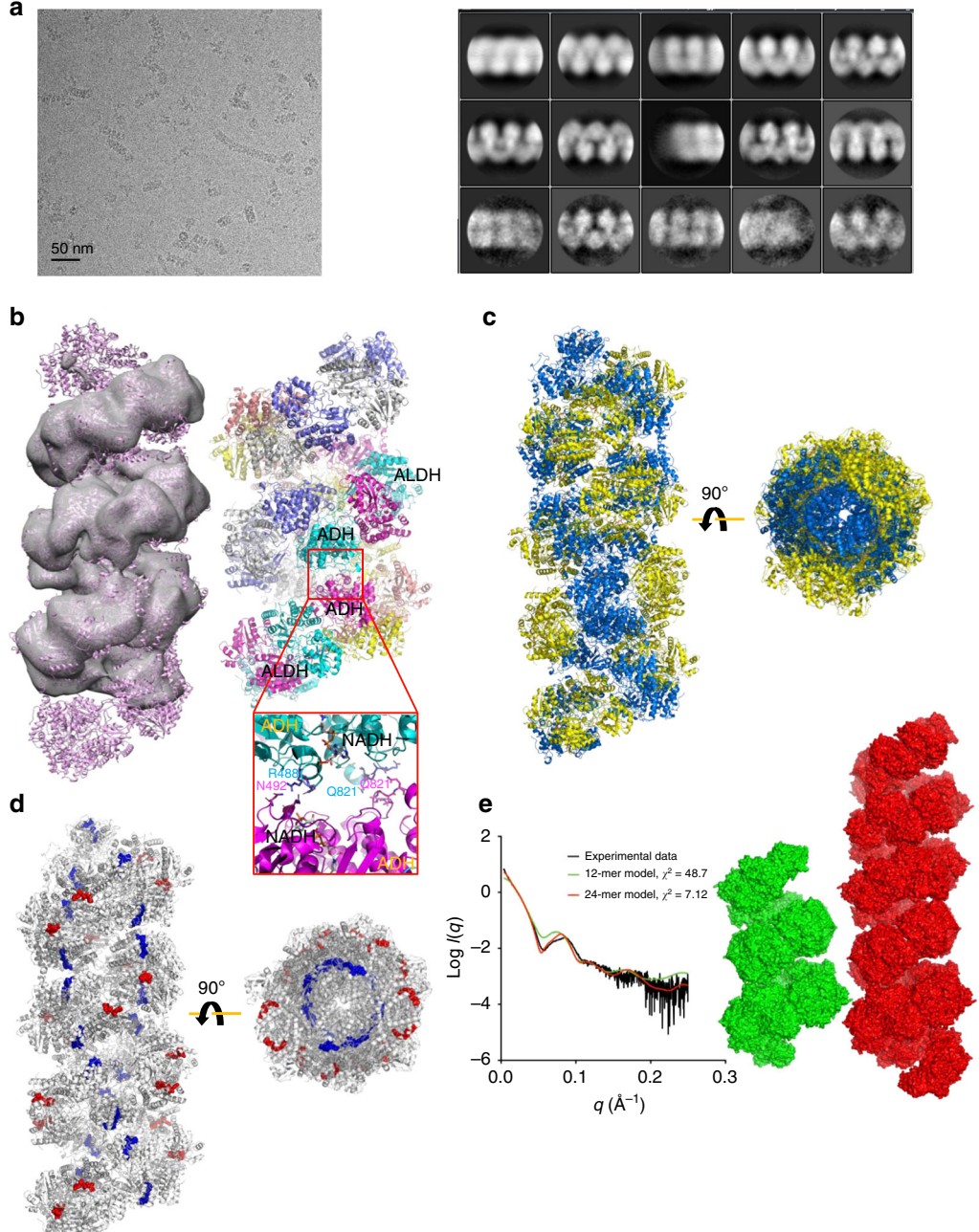

**Fig. 4** Helical reconstruction of AdhE spirosome. **a** A representative micrograph for helical reconstruction (left) and 2D class averages from the helical reconstruction in Relion (right). **b** The high-resolution structures of 12 AdhE molecules were placed on the helical cryo-EM structure (right). Two-and-a-half helical turns of AdhE are shown with the atomic model. The red square indicates the region of the inter-helical interactions between ADH domains. Residues (R488, N492, and Q821) involved in inter-helical interactions are shown in stick representation. **c** ALDH and ADH domains are coloured yellow and blue respectively, revealing domain clustering (side view on left and top view on right). **d** The locations of NAD+ cofactors modeled are shown in space-fill representation, revealing that the ALDH (red) and ADH (blue) catalytic pockets are topologically separated (side view on left and top view on right). **e** Surface representation of 12 (green) and 24 (red) AdhE molecules in spirosome formation (right) for which SAXS profiles (left, green and red lines, respectively), computed with the FoXS server[18,19], fit the experimental SAXS data (left, black line) with $\chi^2$ values of 48.7 and 7.12, respectively

distribution analysis (Supplementary Fig. 9b, c) gave an $R_g$ and $D_{max}$ of 48.4 Å and 205 Å, respectively. The molecular weight was estimated (using SAXSMoW[22]) to be 202.1 kDa, suggestive of a dimeric species (dimeric AdhE would have a molecular weight of 192.2 kDa). Two subunits of AdhE monomer extracted from the high-resolution structure of AdhE were used as input for SOMO and a sedimentation coefficient of 8.5 S was computed for the head-to-head arm-crossing dimer. The discrepancy between this

and that for peaks '2' and '3' suggested that the conformation of "free" dimeric species may differ from that observed within the constraints of the spirosome. Accordingly, the flexibility of the AdhE dimer high-resolution model extracted from the spirosome structure was estimated via normal mode analysis using SREFLEX[23] to generate conformers of the AdhE dimer that best fit the F670E SEC-SAXS data (Fig. 6e). The sedimentation coefficient of the best SREFLEX dimeric model is 8.0 S, in much

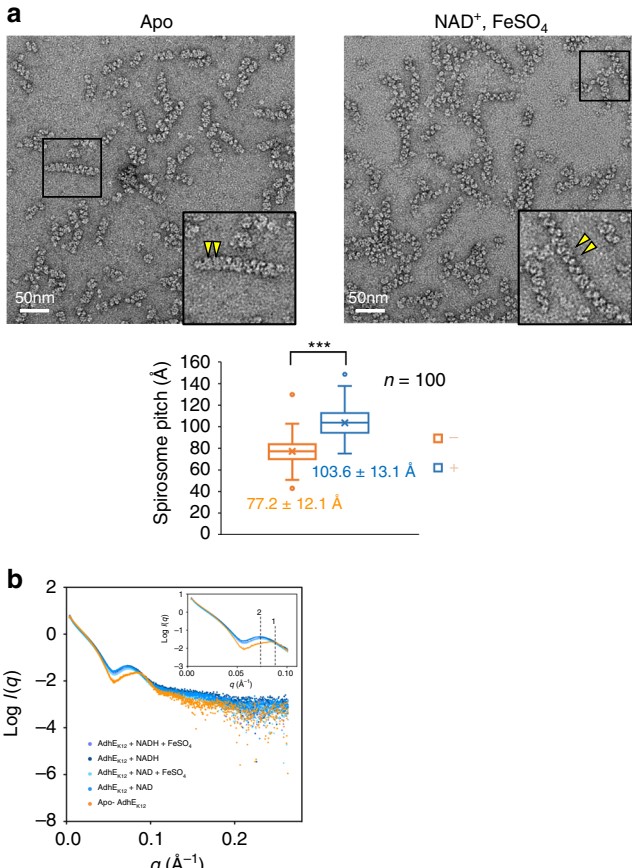

**Fig. 5** AdhE spirosomes change conformation in the presence of cofactors. **a** Negative staining of AdhE in the absence (Apo) and the presence of $NAD^+$ and $FeSO_4$. The yellow triangles indicate the positions of one pitch of the spirosome. The box plot below shows the distribution of spirosome pitch sizes in the absence (- orange, average $= 77.2 \pm 12.1$ Å) and the presence (+ blue, average $= 103.6 \pm 13.1$ Å) of the cofactors ($n = 100$, ***p value: $4.79 \times 10^{-25}$). The box includes the inter-quartile range from Q1 to Q2 and the x in the box indicates the average. The upper and lower dots indicate the maximum and minimum value respectively. **b** SAXS data acquired in batch mode for AdhE fraction 1 in the presence (blue) and absence (orange) of cofactors reveal conformational changes evidenced by a shift in a conserved feature from $q = 0.086$ Å$^{-1}$ (indicated by 1 in the inset) for Apo-AdhE to $q = 0.075$ Å$^{-1}$ (2 in the inset) for AdhE + cofactors, which translates to 73.0 and 83.7 Å in real space

better agreement with the experimentally observed values, suggesting that free AdhE dimer in solution adopts a more extended conformation than within the spirosome. The SREFLEX model along with the SEC-SAXS data for F670E has been deposited in the SASBDB (ID: SASDGN2)[24]. Having generated mutants in which spirosome formation is disrupted, we examined whether the spirosome formation is implicated in AdhE activity. To measure the forward reaction, we incubated AdhE with acetyl-CoA in the presence of NADH and measured the AdhE activity by monitoring the consumption of NADH at 340 nm. Compared with WT, a dramatic decrease in the AdhE activity was observed for the mutants defective in spirosome formation (Fig. 6f), implying that the spirosome formation that we observed in the cryo-EM structure is important for AdhE activity. To examine if the spirosome formation is critical for the reverse reaction to produce acetyl-CoA from alcohol by reducing $NAD^+$ to NADH, we incubated AdhE with ethanol, $NAD^+$ and CoASH, and measured the activity. Interestingly, while the mutations greatly

affected the forward reaction, the reverse reaction was only marginally affected. These data imply that spirosome formation might be more critical for the forward activity of AdhE than its reverse activity. AdhE is a bidirectional and bifunctional enzyme, which converts acetyl-CoA to ethanol via producing acetaldehyde or from ethanol to acetyl-CoA, using two catalytic domains: ALDH and ADH. To investigate further in which step the spirosome formation plays a critical role, we measured the individual forward and reverse activity of ALDH and ADH (Supplementary Fig. 10). As ADH activity requires $Fe^{2+}$ and the reverse reaction of ALDH needs CoASH as a cofactor, we were able to separate the two enzymatic activities by omitting the cofactors in the reaction mixture as done with other ADH enzymatic activity studies[4,25]. Interestingly, most of the activities of the mutants except the ALDH aldehyde reductase activity are comparable to WT. Our data show that spirosome formation plays an important role in the activity of AdhE specifically in the forward activity of ALDH.

## Discussion

In a metabolic pathway, it is often found that several enzymes responsible for consecutive reactions are physically linked to improve the efficiency of the reactions[26]. AdhE is a bifunctional enzyme responsible for converting acetyl-CoA into ethanol, and its ALDH and ADH domains are physically linked. Here, we show that AdhE forms a high-order spirosome structure, which is critical for its activities.

The basic unit of the spirosome structure is composed of an AdhE dimer forming a half helical turn. The AdhE dimer is formed in a cross-arm fashion suggesting that the dimer is obligatory. This unit then forms the spirosome structure via a hydrophobic interaction mediated by the ADH domains. Our in-solution analysis, including gel-filtration, SAXS and AUC, indicated that there is significant heterogeneity in AdhE spirosomes. Consistent with this, the negative stain EM analysis of AdhE also showed spirosomes of different lengths (Fig. 1b and Supplementary Fig. 1). At this moment, it is not clear whether the formation of spirosomes is actively regulated. Considering that the hydrophobic interaction majorly contributes to spirosome formation, the concentration of AdhE in the bacterial cell might affect the degree of spirosome formation, although we have not observed any correlation between the length of spirosome and AdhE concentration in vitro. Furthermore, our SAXS data together with negative stain EM showed that there are at least two different conformations of spirosome structure, dependent on the presence of cofactors, suggesting that the conformational change of spirosome structure might be related to AdhE activity.

By forming the spirosome structure, AdhE can benefit from several advantages in its enzymatic reaction. Firstly, our AdhE cryo-EM structure revealed that spirosome formation leads to clustering of ALDH and ADH catalytic domains, which might have an effect similar to concentrating the enzyme. Furthermore, in the spirosome structure, the inter-helical interaction creates a pocket where two ADH catalytic sites face each other, which might also contribute to an efficient catalytic reaction. Secondly, the structural analysis revealed that the ALDH and ADH activities are topologically separated in the spirosome structure. Both ALDH and ADH utilise NADH to reduce the substrate and produce $NAD^+$ as a product. It can be imagined that $NAD^+$ from one enzyme would inhibit the other enzyme. By topologically separating two enzymes, one enzyme activity is not inhibited by the product of the other's enzymatic activity. In addition, as the product of ALDH (acetaldehyde) is cytotoxic, we postulate that a direct consequence of spirosome formation is that the toxic intermediate will not be released but will instead be sequestered

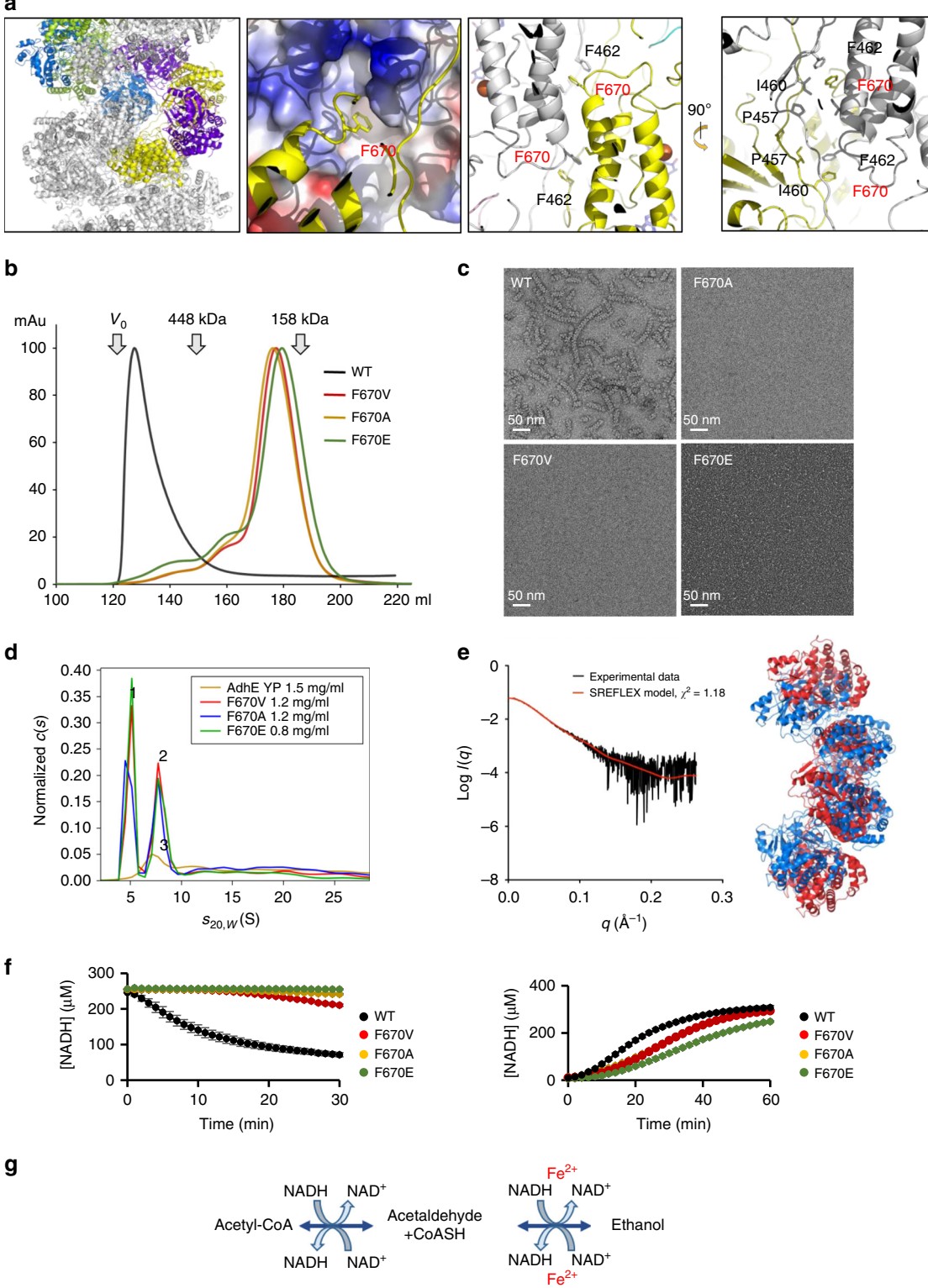

inside the helical structure and further processed (by ADH) to ethanol. Consistent with this, disrupting the spirosome structure affects AdhE activities. Lastly, as ADH resides at the inner surface of the helical structure, a cytotoxic intermediate, acetaldehyde, will be subsequently converted to ethanol without being released into the bacterial cell. Consistent with our observations on the implication of spirosome structure on AdhE activity, our enzymatic analysis with mutants, which cannot form spirosome structures, showed that spirosome formation is indeed critical for

AdhE activity. It is notable that the spirosome formation seems to be more critical for the forward reaction converting acetyl-CoA to ethanol than the reverse reaction consuming ethanol to generate acetyl-CoA. More specifically, the mutations disrupting spirosome formation affect only the forward reaction of ALDH activity, and no other activities such as the reverse reaction of ALDH activity and ADH activity. As the ALDH activity is the first step to reduce acetyl-CoA to aldehyde, and then to ethanol, the spirosome formation might play a role in regulating the

**Fig. 6** The helical organization of AdhE is critical for its activity. **a** The interface between ADH domains. Yellow, purple, blue and green colour indicates four AdhE molecules comprising one helical pitch. F670 inserted into a hydrophobic pocket is shown in a stick model and electrostatic surface representation (second from left). The detailed interaction around F670 with the hydrophobic pocket formed by I460, F462 and I712 (second from right and orthogonal view, right). **b** Gel-filtration profiles of WT AdhE (WT) and the mutants: $AdhE_{F670A}$ (F670A), $AdhE_{F670V}$ (F670V), and $AdhE_{F670E}$ (F670E). **c** Negative stain EM analysis of WT AdhE and the mutants reveals spirosome disruption. **d** c(s) analysis of SV data for $AdhE_{F670A}$ (F670A), $AdhE_{F670V}$ (F670V) and $AdhE_{F670E}$ (F670E) and AdhE expressed from Yersinia pestis ($AdhE_{YP}$). Spirosome disruption by the F670 mutation is evident in the $s_{20,w}$ range ≈ 4–10 S. The distributions for all three F670 mutants include a peak (1) with $s_{20,w}$ = 5.1 S that is absent from $AdhE_{YP}$. However, peak '2' ($s_{20,w}$ = 7.9 S), observed in all F670 mutants, and peak '3' ($s_{20,w}$ ≈ 7.6 S) for $AdhE_{YP}$ are almost overlapping and consistent with $s_{20,w}$ computed for dimeric AdhE. Primary data and quality of the fits to the data are in Supplementary Fig. 8. **e** Cartoon representation of an AdhE dimer extracted from the cryo-EM structure (blue) superimposed (r.m.s.d. = 16.22 Å) on the SREFLEX output model (red) (right) which fits the experimental SAXS data (left, black line) with a $\chi^2$ value of 1.18 (red line). **f** Enzymatic assay of AdhE WT and its mutants. For the forward reaction (left), AdhE was incubated with acetyl-CoA and NADH and the consumption of NADH was monitored. For reverse reaction (right), AdhE was incubated with ethanol, CoASH, and $NAD^+$ and the amount of NADH generated was monitored. The error bars show standard deviation ($n = 3$). **g** A scheme of the reaction

direction of AdhE activity. However, at this moment, it is not clear why spirosome formation is more critical for the forward reaction of ALDH than others. Further structural and biochemical studies might answer this question. Other metabolic enzymes are also found as filamentous forms, including glutaminase C, $CO_2$ reductase, phosphofructokinase-1 and CTP synthase[27–30]. Although the biological significance of filament formation by these enzymes is not yet clear, it might be a general mechanism to regulate the activities of enzymes involved in metabolic pathways. Overall, this work presents the atomic resolution structure of AdhE in a high-order spirosome form and shows that the spirosome structure is critical for its activity. These data imply that AdhE enzymatic activity is regulated by forming a high-order structure.

## Methods

**Expression and purification of AdhE.** The full-length *adhE* gene from Escherichia coli K12 strain was cloned into a pET28a vector for expression with an N-terminal 6-His tag, followed by a tobacco etch virus (TEV) protease cleavage site (ENLYFQ| G). AdhE was overexpressed in BL21 (DE3) RILP cells by induction with 0.5 mM IPTG at 18 °C for 18 h when the $OD_{600}$ reached 0.7–0.8. Cells were harvested by centrifugation at 4000 r.p.m. (4553 × g) for 20 min The cell pellets were resuspended and sonicated in a lysis buffer (buffer A) containing 50 mM Tris-HCl pH 8.0, 500 mM NaCl and 5% (v/v) glycerol. The cell lysate was clarified by centrifugation at 18,000 r.p.m. (39,204 × g) for 1 h, and supernatant was incubated with Ni-NTA agarose beads (Qiagen). The Ni-NTA beads were washed with buffer A containing 20 mM imidazole, and bound protein was eluted with buffer A containing 200 mM imidazole. The N-terminal His-tag was cleaved by TEV protease during an overnight dialysis step against buffer containing 50 mM Tris-HCl pH 8.0, 100 mM NaCl, 1 mM DTT and 0.5 mM EDTA at 4 °C. AdhE was further purified on a HiTrap Heparin HP (GE Healthcare) ion exchange chromatography column and a Superdex 200 (GE Healthcare) size exclusion chromatography column equilibrated with buffer containing 50 mM Tris-HCl pH 8.0, 500 mM NaCl and 1 mM DTT. Fractions containing AdhE were concentrated with Amicon Ultra 30,000 MWCO centrifugal filters (Millipore) up to 5 mg/ml and flash frozen in liquid nitrogen before storage at −80 °C. AdhE mutants (F670E, F670A, and F670V) were generated with a QuikChange site-directed mutagenesis kit (Stratagene) and purified similarly to wild-type AdhE. The sequences of primers used in this study are listed in Supplementary Table 4.

**Negative stain EM**. 3 μl drops of purified AdhE (0.05 mg/ml) were applied to glow-discharged carbon coated Cu (400 mesh) grids and incubated for 1 min The grids were washed twice with water and were incubated with 1% (w/v) uranyl acetate for 1 min for negative staining. Excess uranyl acetate was removed by filter paper and the grids dried. Prepared grids were analysed using a Tecnai F20 electron microscope (FEI) with CCD camera (Gatan).

**Cryo-EM sample preparation and image processing.** 3 μl drops of purified AdhE (5.1 mg/ml) were applied to glow-discharged R2/2 Quantifoil holey grids (200 mesh). The protein was blotted for 3 s with −10 blotting force for 100% humidity and plunge-frozen using a Vitrobot Mark IV (ThermoFisher Scientific). Micrograph images in movie mode were collected using a Titan Krios (Thermo-Fisher Scientific) with a Falcon III direct detector at the Korea Basic Science Institute (KBSI), Ochang, Korea operated at 300 keV, 1.12 Å/pixel, 44.6 e/Å²/ micrograph with a −0.5 to −3.0 μm defocus range and 80 movie frames. Thirty-nine movie frames from the second frame onwards (total dose 21.8 e/Å²) were aligned with the program cisTEM[14] and all the following images were processed

with cisTEM. A total of 251,604 particles were initially picked and 160,830 particles were finally selected from good 2D class averages after several rounds of the 2D class averaging process. Using the selected particles, an initial 3D model was generated and further refined with the auto refine mode without any symmetry. The model was built with the program COOT[31] and refined with the real-space refinement procedure implemented in the program PHENIX[32]. 92.2% of amino acids in full-length AdhE were unambiguously registered in the cryo-EM density.

For the helical reconstruction process, 3 μl drops of fraction 1 (0.8 mg/ml) containing helical structures were applied to glow discharged R2/2 Quantifoil holey grid (300 mesh) grids and blotted for 2 s with −15 blotting force, then plunge-frozen using a Vitrobot Mark IV (ThermoFisher Scientific). Micrographs in integration mode were collected using a Talos Artica (ThermoFisher Scientific) with a Falcon III direct detector at the SciLifeLab, Stockholm, Sweden operating at 200 keV, 2.02 Å/pixel, 50 e/Å²/micrograph with a −1 to −2.5 μm defocus range. All frames, except the first two, were aligned with MotionCor2[33], and CTF correction was performed using Gctf implemented in Relion[17]. The cryo-EM map and resultant coordinates were deposited in the EMDataBank[34] (ID:EMD-9623) and PDB[35] (PDB ID: 6AHC) respectively. For the helical reconstruction process, the particles at the start and end positions in the helix were manually picked and a total of 43,151 particles were picked in a helical picking mode, and subsequently processed for 3D reconstruction in helical mode with parameters of initial rise 18 Å, initial twist 90°, central Z length 25%, inner tube diameter 10 Å and outer tube diameter 150 Å.

**Enzymatic activity assay**. To determine the enzymatic activity of the WT and F670 mutant AdhE, the consumption or production of NADH was measured at a wavelength of 340 nm using a Synergy H1 Microplate Reader (BioTek). All assays were performed at 37 °C and the total volume was 100 μl. WT and F670 mutant AdhE were prepared in 50 mM Tris-HCl pH 8.0. The activities of the AdhE forward reaction combined with ALDH and ADH were measured in a reductase activity assay mixture containing 50 mM Tris-HCl pH 8.0, 20 μM $FeSO_4$, 200 μM acetyl-CoA, and 250 μM NADH with 6 μg (0.06 nmol) of AdhE and the consumption of NADH immediately monitored. For measuring the acetyl CoA reductase activity (ALDH forward reaction), the assays were performed in reaction mixtures containing 50 mM Tris-HCl pH 8.0, 20 μM $FeSO_4$, 200 μM acetyl-CoA, and 250 μM NADH without $FeSO_4$ and with 20 μM EDTA pH 8.0 to chelate Fe, which is required for ADH, with 17.6 μg of AdhE. For measuring acetaldehyde reductase activity (ADH forward reaction), the assays were performed in reaction mixtures containing 50 mM Tris-HCl pH 8.0, 20 μM $FeSO_4$, 100 μM acetaldehyde, and 250 μM NADH with 6 μg of AdhE. The activities of the AdhE reverse reaction were performed in reaction mixtures containing 50 mM Tris-HCl pH 8.0, 20 μM $FeSO_4$, 200 μM CoA-SH, 200 mM ethanol, and 500 μM $NAD^+$. With 22 μg (0.22 nmol) of AdhE and the production of NADH immediately monitored at 340 nm. For measuring alcohol dehydrogenase activity (ADH reverse reaction), the assays were perfomed in reaction mixtures containing 50 mM Tris-HCl pH 8.0, 20 μM $FeSO_4$, 200 μM CoA-SH, 200 mM ethanol, and 500 μM $NAD^+$ with 22 μg of AdhE. To measure only ADH dehydrogenase activity, 200 μM CoA-SH was removed from the ethanol dehydrogenase assay mixture. The acetaldehyde dehydrogenase activity (ALDH reverse reaction) was measured in a reaction mixture containing 50 mM Tris-HCl pH 8.0, 20 μM $FeSO_4$, 200 μM CoA-SH, 100 mM acetaldehyde, and 500 μM $NAD^+$ with 22 μg (0.22 nmol) of AdhE and the production of NADH immediately monitored at 340 nm.

**Analytical ultracentrifugation (AUC)**. Sedimentation velocity (SV) experiments were performed using a Beckman Coulter XL-I analytical ultracentrifuge equipped with an An-50 Ti eight-hole rotor. 300–360 μl of samples were loaded into 12 mm pathlength charcoal-filled epon double-sector centrepieces, sandwiched between two sapphire windows and equilibrated at 4 °C in vacuum for 6 h before running at 49 k rpm. The laser delay, brightness, and contrast were pre-adjusted at 3 k rpm to acquire the best quality interference fringes. Data were

collected using Rayleigh interference and absorbance optics recording radial intensity or absorbance at 280 nm between radial positions of 5.65 and 7.25 cm, with a radial resolution of 0.005 cm and a time interval of 7 min, and analysed with the program SEDFIT[36] using a continuous c(s) model. The partial specific volume, buffer density and viscosity were calculated using SEDNTERP[37] (Supplementary Table 2).

**Small angle-X-ray scattering (SAXS).** SAXS was done on beamline B21 of the Diamond Light Source synchrotron facility (Didcot, UK). Data were recorded at 12.4 keV, at a sample-detector distance of 4.014 m using a Pilatus 2 M detector (Dectris, Switzerland). For batch mode measurements, samples (30 μl at concentrations between 3.8 and 5.2 mg/ml) and solvent were loaded into a 96-well plate, before being sequentially injected into a quartz capillary by the BioSAXS robot. For SEC-SAXS 50 μl of protein samples at concentrations of 9–10 mg/ml were loaded onto either a Shodex KW-404 (for MW < 100 kDa) or a Shodex KW-405 (for MW > 100 kDa) size exclusion chromatography column (Showa Denko, Japan) in 50 mM HEPES pH 7, 500 mM NaCl, 5% (v/v) glycerol at 0.16 ml/min using an Agilent 1200 HPLC system. The column outlet was fed into the experimental cell, and $620 \times 3.0$ s frames of SAXS data were recorded. Data were processed with ScÅtter (http://www.bioisis.net) as follows. The integral of ratio to background signal along with the estimated radius of gyration (Rg) for each frame was plotted. Frames within regions of low signal and low Rg were selected as buffer and subtracted from frames within regions of higher signal and constant Rg. Subsequent SAXS analysis was performed using the ATSAS 2.8 suite of programs[23]. The radius of gyration Rg was obtained from the Guinier approximation[38] following standard procedures. The pairwise distance distribution function P(r) was computed using the indirect Fourier transformation method implemented in GNOM[39]. From the P(r) function, alternative estimates of Rg and maximum particle dimension $D_{max}$ were obtained. All SAXS data and parameters are listed in Supplementary Table 3.

**Reporting summary**. Further information on research design is available in the Nature Research Reporting Summary linked to this article.

## Data availability

The cryo-EM density maps and the atomic coordinates have been deposited in the Electron Microscopy Data Bank (EMDB) and in the PDB, respectively, under the following accession codes: EMD-9623 and 6AHC [https://www.rcsb.org/structure/6AHC]. The source data underlying Figs. 5a, 6f and Supplementary. Figs 1b, 1c, 9b and 10 are provided as a Source Data file. Other data are available from the corresponding author upon reasonable request.

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

## Acknowledgements

We thank the staff at the Korea Basic Science Institute (KBSI) and the staff at the SciLifeLab for help with cryo-EM data collection. We are grateful to Diamond Light Source for the SAXS beamtime and to the B21 beamline scientists for excellent scientific support. We also thank Dr. Carol Cho for critical reading. We also thank Donghui Choe and Kangsan Kim for helping with the enzymatic assays. Computing resource was supported by Global Science experimental Data hub Center (GSDC), Korea Institute of Science and Technology Information (KISTI). This work is partially supported by grants (NRF-2016R1A2B3006293, NRF-2016K1A1A2912057. GSDC-NRF-2018R1A6A7052113 to J.S.) from the National Research Foundation of Korea, the Grand Challenge 30 program (to J.S.) from KAIST. This work was partially supported by the Intelligent Synthetic Biology Center (ISBC) of Global Frontier Project funded by the Ministry of Science and ICT(MSIT) (2011-003955). G.K. is a recipient of the Global Fellowship (NRF-2018H1A2A1061362) and L.B.A. is a recipient of the Skim Latihan Bumiputera from the Ministry of Higher Education Malaysia and Universiti Sains Islam Malaysia.

## Author contributions

G.K., S.J., and J.S. conceived the idea. G.K., T.J., H.H., and J.S. performed the cryo-EM study, and G.K. and J.S. performed in vitro assay. L.B.A., A.R., and O.B. performed AUC and SAXS experiments. All authors examined the data and wrote the manuscript.

## Competing interests

The authors declare no competing interests.
