## [Peer Review File · Nature Communications]

Reviewers' comments:

Reviewer #1 (Remarks to the Author):

Kim et al. represent the structure of Aldehyde-Alcohol dehydrogenase and its ability to form higher order structures in the shape of spiroosomes. They have used electron cryo microscopy, and single particle image reconstruction to determine the structure to some 3.5 Å resolution. They propose that oligomerization could impact on function. To test this hypothesis the authors investigated the structural impact of mutations and cofactors onto the spiroosome structure and checked how activity changes in mutants with different oligomerization states. Many of these structural aspects are tested by SAXS or by negative staining electron microscopy. It is somewhat disappointing that no high-resolution EM-structures are offered for comparison. In particular the structural changes on the addition of co-factors could have been better described with high-resolution cryo-EM. However, by and large, this is an interesting study, which enhances our understanding of switching activity of complexes by oligomerization. The study is sound and tests the findings with many different methods.

Specific remarks:

- 1) Page 5 spiroosomes: How long do spiroosomes grow? What prevents them from growing longer? Is there anything in the structure of the spiroosomes that hints a helical mismatch and therefore could explain, why the spiroosomes do not grow indefinitely?
- 2) Line 94: "3.45Å" – Please quote only 2 digits > "3.5Å". The determination of resolution is not that accurate
- 3) Line 100-101: Please quote the helical rise and helical rotation (as would be used for helical refinement)
- 4) Line 117 "... which might be important for its biological activity": Please explain in which respect. Be a bit more specific here (e.g. Inhibitory?, activating?) Similar, general phrases are found throughout the text (e.g. line 151) and should be rephrased more meaningful.
- 5) Line 128: This reviewer finds the 11.2 Å resolution disappointing considering the equipment and the amount of data. Please explain what is preventing higher resolution here. Are the helices more variable? Do they have deviations from helical symmetry, which could explain why they are not growing longer?
- 6) Figure4: The fit shows many parts of the model, which stick out of the map. Can the authors rule out some rearrangement of the fold of the monomer? Where does the confident part of the helical map start and end?
- 7) Line 164-165 – "... extended along the long axis": How does this extension with co-factors compare to classes 2 and 3 of the spiroosome without co-factors, which represent the majority of the particles? These two classes appear more open and more extended than the best resolved class that was further analysed. Therefore, it is unclear whether co-factors trigger a conformational change or whether such an extension is already present in the data of the particles without co-factors.
- 8) Figure 5B, why is this type of comparison not done in cryo-EM, which would give a much more direct insight into the structural differences than SAX-scattering curves? In particular SAXS curves become hard to interpret if several conformational states are present as suggested by the classes of the spiroosome shown in S3.
- 9) Figure 5A does not look very representative. How do other filaments look like? Make a gallery of many typical filaments for the two species.
- 10) Figure 6C: Please enlarge the micrographs to make the monomers visible. F670A and F670V are not properly stained (positive staining), please show negatively stained areas only!!!
- 11) Line 210 dimers: Why don't you do cryo EM on the dimers, to confirm the dimer structure and to show whether the dimeric building block of the mutants has the same structure as in the spiroosomes? With 200 kDa MW the size should be sufficient for cryo-EM

Bettina Böttcher

Reviewer #2 (Remarks to the Author):

Filamentous form of metabolic enzymes is a fascinating phenomenon that has not attracted much attention. Neither the type of assembly or the structure of the filaments is known nor whether filament formation is of physiological or kinetic importance. In light of this, the study by Kim et al is very much appreciated. This reviewer is not a specialist in structural biology and can not judge whether the structures presented are justified by the original data. However, the presented structure and its interpretation with respect to a separation of enzymatic activities of the bifunctional enzyme is extremely interesting and appealing to a broad audience. This is the first study that revealed the structure of the filaments as well as gives first insights into the function of filament formation by enzymatic analyses followed site directed mutagenesis to interrupt the two activities of the enzyme.

The paper would benefit from the following:

L51: must be rephrased: there is no such thing as „acetobacteria“. Since the authors referenced Bertsch et al, they probably mean one species, *Acetobacterium*. Rephrasing such as: "conserved amongst anaerobic bacteria such as primary fermenters (enterobacteria, clostridia) and acetogenic bacteria such as *Acetobacterium woodii*".

Results: AdhE is a bifunctional enzyme that converts acetyl-CoA via acetaldehyde to ethanol. It also can use acetaldehyde as a substrate and oxidize it to acetyl-CoA and it can also reduce acetaldehyde to ethanol. Unfortunately, these partial activities have not been determined by the authors although this is easy to do and would strengthen the manuscript considerably. Especially in the context of the mutation and the observation that only the forward direction is affected in the variant.

Please be careful with the terms „bidirectional“ and „bifunctional“. Any enzyme is bidirectional since it only catalyzes reaching the equilibrium. Bifunctional means two different reactions.

Discussion: would benefit from mentioning other filamentous forms of metabolic enzymes: acetyl-CoA carboxylase, CTP synthase and last but not least, the youngest member of this class, the hydrogen dependent CO₂ reductase of *Acetobacterium woodii*, for which filament formation and its physiological role was described rather recently.

Reviewer #3 (Remarks to the Author):

The CryoEM structure looks very pretty, and it appears to correlate with the activity of the enzyme.

While CryoEM is not my field, the reporting of the data collection and the analysis thereof meets with what I have observed for other CryoEM studies.

SAXS and AUC are my fields, and this is where I have a few concerns. I suggest that the authors refer to Prof. Jill Trewhell's guidelines on the reporting of SAXS data, as this is becoming a standard for the small angle scattering community.

-page 8, line 147: The SAXS profile shows the reciprocal space position as $q = 0.075$ and χ^2 value of 7.12" makes no real sense. I am not clear what the authors mean by a "reciprocal space position" and SAXS data does not have an inherent χ^2 value – χ^2 is a measure of how well a model represents the data, and this sentence does not refer to a fit of a model. On inspection of figure 4e, this χ^2 appears to refer to the fit of the 24mer spiroosome to the data.

-The next sentence "This SAXS profile is fitted well with the computed SAXS profile from a model of the spiroosome composed of 24 AdhE molecules calculated using the FoXS server" is not supported by the data. A χ^2 value of 7.12 means that the fit does not represent the scattering profile. This may be due to a number of factors like flexibility, differing arrangements, or polydispersity, and probably should be explored further.

-In particular, I am not confident that the data in figure 4e is from a monodisperse population, particularly as there is no concentration dependence presented for the scattering profiles, no

Guinier plot of this data, no $P(r)$ for this profile, and no estimation of MW (which would lend support to the 24mer). These parameters are almost always required for the validation of protein scattering data, and should be included in supplemental data. Parameters from a SEC SAXS run cannot be used to support a batch SAXS measurement, and each dataset should be able to be verified. Furthermore, inspection of the low q data in figure 4e suggests that presence of significantly longer species, and appears to indicate a non-monodisperse solution, which would be consistent with the SEC and AUC analyses presented elsewhere in the manuscript. If this is the case, then it is not possible to fit a structure to this data, as monodispersity is an absolute requirement for the modeling of protein SAXS data, be it with existing high resolution structures or ab initio.

-Fig 5; while I agree that there is a conformational change, I don't necessarily agree that the feature correlates directly to a real space change from ~80 angstrom to ~70 angstrom as asserted on page 9, line 175. SAXS profiles are the Fourier transform of all distances inside a particle, and features such as these can arise from form factors (spheres, cylinders etc...) in a way that is related to the overall size of particle. This is not diffraction data, where the peak position arises from d-spacing in electron density, and without a full analytical model of the data no conclusion can really be drawn about what real space physical distance(s) give rise to this feature. Further, the number of curves in panel b obscures the differences in the data, and really makes panel b not very informative nor supportive of the conclusions drawn by the authors.

-As an aside, the ApoAdhEk12 data (the orange curve) looks to my eye to be somewhat over subtracted. Buffer matching would be better supported if a concentration dependence for these data were provided in the supplemental data. It is conceivable that the large change in comparison the other curves could be caused by a poor background subtraction.

-Fig 6. The AUC data, in the form of the $c(s)$ distribution looks wrong. There's either too few points in the $c(s)$ distribution, or the regularization has not worked as there should be no sharp, jagged, peaks in a $c(s)$ distribution like the ones labeled 1 and 2. There's no primary data from the AUC runs included in the manuscript (such as the absorbance (or interference) v radial position over time plots) and no evidence of the quality of the $c(s)$ model fit. The authors don't quote a frictional ratio, which has a significant impact on the data. Was this parameter fixed for all samples? Was it allowed to float? How was the impact of large, very slowly diffusing particles on the smaller faster diffusing particles taken into account?

On the whole, the manuscript was well written; and with proper reporting and careful analysis of the SAXS data I would be amenable to recommending it for publication.

Response to the reviewers' comments

Reviewers' comments:

Reviewer #1 (Remarks to the Author):

Kim et al. represent the structure of Aldehyde-Alcohol dehydrogenase and its ability to form higher order structures in the shape of spiroosomes. They have used electron cryo microscopy, and single particle image reconstruction to determine the structure to some 3.5 Å resolution. They propose that oligomerization could impact on function. To test this hypothesis the authors investigated the structural impact of mutations and cofactors onto the spiroosome structure and checked how activity changes in mutants with different oligomerization states. Many of these structural aspects are tested by SAXS or by negative staining electron microscopy. It is somewhat disappointing that no high-resolution EM-structures are offered for comparison. In particular the structural changes on the addition of co-factors could have been better described with high-resolution cryo-EM. However, by and large, this is an interesting study, which enhances our understanding of switching activity of complexes by oligomerization. The study is sound and tests the findings with many different methods.

Specific remarks:

1) Page 5 spiroosomes: How long do spiroosomes grow? What prevents them from growing longer? Is there anything in the structure of the spiroosomes that hints a helical mismatch and therefore could explain, why the spiroosomes do not grow indefinitely?

→ In the revised manuscript, we analyzed the length distribution of spiroosomes. The length varies from 150 to 1200 Å, and the distribution is also quite diverse, as shown in Supplementary Fig. 1. At this moment, it is not clear whether the length of spiroosomes is regulated nor, if so, how such regulation is achieved. Despite the high resolution structure, we could not find any hint regarding how the growth of spiroosome can be controlled. As we did not observe any correlation between the length of spiroosomes and the concentrations, the length may not be dependent on concentration. In the revised manuscript, we included a new Supplementary Fig. 1 showing the length distribution of spiroosomes in Fraction 1.

2) Line 94: “3.45Å” – Please quote only 2 digits > “3.5Å”. The determination of resolution is not that accurate

→ Thank you for the comment, we changed the resolution from 3.45Å to 3.5Å throughout the manuscript including Supplementary Table 1, accordingly.

3) Line 100-101: Please quote the helical rise and helical rotation (as would be used for helical refinement)

→ We appreciate the reviewer's comment. We would like to clarify that we solved two cryo-EM structures, one at high-resolution and the other at low resolution. For the high-resolution structure shown in Figs. 1, 2 and 3, we performed single-particle processing using cisTEM with fraction 2 where AdhE forms shorter spiroosomes, as long spiroosomes are not straight and a helical processing did not produce high resolution 3D reconstruction. For the low-resolution spiroosome structure shown in Fig. 4, we performed helical processing implemented in Relion 3. According to the reviewer's comment, we included the parameters used for helical processing in the Methods as ‘...for 3D reconstruction in helical mode with parameters of initial rise 18 Å, initial twist 90°, central Z length 25%, inner tube diameter 10 Å and outer tube diameter 150 Å.’

4) Line 117 “... which might be important for its biological activity”: Please explain in which respect. Be a bit more specific here (e.g. Inhibitory?, activating?) Similar, general phrases are found throughout the text (e.g. line 151) and should be rephrased more meaningful.

→ According to the reviewer’s suggestion, we replaced general phrases with sentences as follow
Line 117: “...which might lead to activating its biochemical activity by clustering enzymes.’
Line 151: ‘...by which the activity of AdhE might be activated.’

5) Line 128: This reviewer finds the 11.2 Å resolution disappointing considering the equipment and the amount of data. Please explain what is preventing higher resolution here. Are the helices more variable? Do they have deviations from helical symmetry, which could explain why they are not growing longer?

→ As indicated in Supplementary Table 1 and Method, the data for long-spirosomes were collected with Talos Arctica in an integration mode, which has low DQE. Collecting data in an integration mode might prevent the attainment of high resolution. In addition, we initially performed helical processing with the data from fraction 2, which generated high-resolution structure by single-particle process. However, the helical processing did not give high resolution 3D reconstruction as the single-particle processing did. We noticed that the spirosomes are not straight and this also might affect the resolution during helical processing. As we described in 1), it is not clear why the spirosomes are not infinitely growing and we could not get any hint from even the high-resolution structure. Despite the low resolution structure of the long-spirosome, the current high-resolution structure contains one-and-half helical turn, which allowed us to reconstitute long-spirosome. In addition, the reconstituted long-spirosome matches well with the low-resolution 3D reconstruction resulted from helical processing shown in Figure 4b.

6) Figure4: The fit shows many parts of the model, which stick out of the map. Can the authors rule out some rearrangement of the fold of the monomer? Where does the confident part of the helical map start and end?

→ The reason why the model sticks out the map shown in Figure 4 is due to the parameters such as box size (505 Å) and mask size (400 Å) set during the helical processing, not due to the fact that the actual structure stops. The reason that the model sticks out of the map is because the map was generated with a certain maximum setting of the helical length during the helical processing not because of some rearrangement of the monomer.

7) Line 164-165 – “... extended along the long axis”: How does this extension with co-factors compare to classes 2 and 3 of the spirosome without co-factors, which represent the majority of the particles? These two classes appear more open and more extended than the best resolved class that was further analysed. Therefore, it is unclear whether co-factors trigger a conformational change or whether such an extension is already present in the data of the particles without co-factors.

→ The reviewer mentioned classes 2 and 3, and we assume that the reviewer meant Fractions 2 and 3 in Fig. 1. As we described in the main text in Page 5, the high-resolution cryo-EM structure was determined using Fraction 2 and the low-resolution structure by helical processing was done using Fraction 1. Fig. 4a shows that the pitches of both structures are identical, indicating that the pitches of AdhE in Fraction 2 and 3 are same. Fraction 3 seems to contain the dimer of AdhE which does not have a pitch. In the revised manuscript, we statistically analyzed the length of the pitch in the absence and presence of the co-factor shown (new Fig. 5a). The average pitch of the spirosome significantly increases upon addition of the co-factor, suggesting that the co-factor might induce this change. As we were able to observe some spirosomes in an extended conformation even without the co-factors, some spirosomes in an extended conformation present even without co-factors. However, our analysis of 100 molecules together with

SAXS data suggests that the spiroosome adopts more extended conformations in the presence of the co-factors. In order to tone down our assertions, we replace a sentence ‘...suggesting that cofactor binding might induce substantial conformational change within the spiroosome structure.’ with ‘...suggesting that spiroosomes in an extended conformation are formed upon addition of the cofactors.’

8) Figure 5B, why is this type of comparison not done in cryo-EM, which would give a much more direct insight into the structural differences than SAX-scattering curves? In particular SAXS curves become hard to interpret if several conformational states are present as suggested by the classes of the spiroosome shown in S3.

→ It would indeed be a much more direct comparison if we could obtain the cryo-EM structure of AdhE in an extended conformation. Despite our attempts in this regard, we were not able to find a good grid condition for AdhE in the presence of the co-factors. To further clarify the difference of the conformation of AdhE in the absence and presence of the cofactors, we further analyzed the difference between two species by measuring the pitches of AdhE – the results are shown in a graph in Fig. 5a.

9) Figure 5A does not look very representative. How do other filaments look like? Make a gallery of many typical filaments for the two species.

→ According to the reviewer’s suggestion, new micrographs with a large field of view showing many filaments are included in Figure 5a. In addition, we statistically analyzed the pitch difference between the two spiroosome species – these results are shown in a graph in Fig. 5a.

10) Figure 6C: Please enlarge the micrographs to make the monomers visible. F670A and F670V are not properly stained (positive staining), please show negatively stained areas only!!!

→ We repeatedly examined the F670A and F670V mutants by negative-staining. However, we were not able to have clear negative staining as for F670E. However, we believe that this is not due to the quality of the samples. All mutants gave symmetric chromatograms at the elution position of corresponding to a molecular weight of 180 kDa, indicating that the point mutations did not disrupt the global structure of AdhE. In addition, the enzymatic activities of all mutants, converting ethanol to acetyl-CoA (reverse reaction) are comparable to that of the WT, further verifying that the mutants maintain intact structures. Collectively, the staining issue in the mutants is not because the samples are degraded or misfolded. In the revised manuscript, we replaced with enlarged images.

11) Line 210 dimers: Why don't you do cryo EM on the dimers, to confirm the dimer structure and to show whether the dimeric building block of the mutants has the same structure as in in the spiroosomes? With 200 kDa MW the size should be sufficient for cryo-EM.

→ It would be interesting to compare the structures of the dimer, and we actually tried to pursue to this direction. However, it was not straightforward as cryo-conditions seem to be totally different from those required for cryo-imaging of the spiroosome. Cryo-EM study on the dimer would be another new project. As described in 10), all mutants gave symmetric chromatograms at the elution position of corresponding to a molecular weight of 180 kDa, indicating that the point mutations did not disrupt the global structure of AdhE although the relative orientation between two domains might differ.

Reviewer #2 (Remarks to the Author):

Filamentous form of metabolic enzymes is a fascinating phenomenon that has not attracted much attention. Neither the type of assembly or the structure of the filaments is known nor whether filament formation is of physiological or kinetic importance. In light of this, the study by Kim et al is very much appreciated. This reviewer is not a specialist in structural biology and can not judge whether the structures presented are justified by the original data. However, the presented structure and its interpretation with respect to a separation of enzymatic activities of the bifunctional enzyme is extremely interesting and appealing to a broad audience. This is the first study that revealed the structure of the filaments as well as gives first insights into the function of filament formation by enzymatic analyses followed site directed mutagenesis to interrupt the two activities of the enzyme.

The paper would benefit from the following:

L51: must be rephrased: there is no such thing as „acetobacteria“. Since the authors referenced Bertsch et al, they probably mean one species, *Acetobacterium*. Rephrasing such as: "conserved amongst anaerobic bacteria such as primary fermenters (enterobacteria, clostridia) and acetogenic bacteria such as *Acetobacterium woodii*".

→ We thank the reviewer for pointing this out. We rephrased the sentence as suggested by the reviewer.

Results: AdhE is a bifunctional enzyme that converts acetyl-CoA via acetaldehyde to ethanol. It also can use acetaldehyde as a substrate and oxidize it to acetyl-CoA and it can also reduce acetaldehyde to ethanol. Unfortunately, these partial activities have not been determined by the authors although this is easy to do and would strengthen the manuscript considerably. Especially in the context of the mutation and the observation that only the forward direction is affected in the variant.

→ As the reviewer suggested, we measured the partial activities. The data indicate that the mutants seem to have a defect only in the forward reaction of ALDH and all other activities are comparable to WT. We discuss these results in the Result section and included the data as Supplementary Fig. 10 in the revised manuscript. We appreciate the reviewer's comments on this.

Please be careful with the terms „bidirectional“ and „bifunctional“. Any enzyme is bidirectional since it only catalyzes reaching the equilibrium. Bifunctional means two different reactions.

→ We appreciate the reviewer's comment on this. We used 'bidirectional' when we mean AdhE can convert acetyl CoA to ethanol, and ethanol to acetyl CoA. We used 'bifunctional' when we mean AdhE has two catalytic activities: ALDH and ADH. We double checked to make sure that there is no confusion in these terms throughout the manuscript.

Discussion: would benefit from mentioning other filamentous forms of metabolic enzymes: acetyl-CoA carboxylase, CTP synthase and last but not least, the youngest member of this class, the hydrogen dependent CO₂ reductase of *Acetobacterium woodii*, for which filament formation and its physiological role was described rather recently.

→ As the reviewer suggested, we mentioned other filamentous forms of enzymes in the discussion with proper references.

Reviewer #3 (Remarks to the Author):

The CryoEM structure looks very pretty, and it appears to correlate with the activity of the enzyme. While CryoEM is not my field, the reporting of the data collection and the analysis thereof meets with what I have observed for other CryoEM studies.

SAXS and AUC are my fields, and this is where I have a few concerns. I suggest that the authors refer to Prof. Jill Trehwell's guidelines on the reporting of SAXS data, as this is becoming a standard for the small angle scattering community.

→ According to the reviewer suggestion, we included SAXS data collection and parameters as Supplementary Table 3.

-page 8, line 147: The SAXS profile shows the reciprocal space position as $q = 0.075$ and χ^2 value of 7.12” makes no real sense. I am not clear what the authors mean by a “reciprocal space position” and SAXS data does not have an inherent χ^2 value – χ^2 is a measure of how well a model represents the data, and this sentence does not refer to a fit of a model. On inspection of figure 4e, this χ^2 appears to refer to the fit of the 24mer spiroosome to the data.

→ Addressed together with 1) 2) and 3) in 3). Please refer to our response to 3) below

-The next sentence “This SAXS profile is fitted well with the computed SAXS profile from a model of the spiroosome composed of 24 AdhE molecules calculated using the FoXS server” is not supported by the data. A χ^2 value of 7.12 means that the fit does not represent the scattering profile. This may be due to a number of factors like flexibility, differing arrangements, or polydispersity, and probably should be explored further.

→ Addressed together with 1) 2) and 3) in 3). Please refer to our response to 3) below

-In particular, I am not confident that the data in figure 4e is from a monodisperse population, particularly as there is no concentration dependence presented for the scattering profiles, no Guinier plot of this data, no $P(r)$ for this profile, and no estimation of MW (which would lend support to the 24mer). These parameters are almost always required for the validation of protein scattering data, and should be included in supplemental data. Parameters from a SEC SAXS run cannot be used to support a batch SAXS measurement, and each dataset should be able to be verified. Furthermore, Inspection of the low q data in figure 4e suggests that presence of significantly longer species, and appears to indicate a non-monodisperse solution, which would be consistent with the SEC and AUC analyses presented elsewhere in the manuscript. If this is the case, then it is not possible to fit a structure to this data, as monodispersity is an absolute requirement for the modeling of protein SAXS data, be it with existing high-resolution structures or ab initio.

→ This text on pages 7-8 was erroneously abbreviated in the final stages of editing prior to submission, for which we apologise. The text which originally read

“To further examine the properties of AdhE spiroosome in solution, we applied small angle X-ray scattering (SAXS) method. SAXS data for AdhE was obtained in batch mode without further fractionation by SEC. The SAXS profile shows the reciprocal space position as $q = 0.075$ and χ^2 value of 7.12. This SAXS profile is fitted well with the computed SAXS profile from a model of the spiroosome composed of 24 AdhE molecules calculated using the FoXS server (Fig. 4e). Overall, these data show that AdhE forms into a spiroosome structure, which may be critical for its function.”

Has been changed to read

“To further examine the properties of AdhE spiroosomes in solution, we undertook small angle X-ray scattering. SAXS data for AdhE were obtained in batch mode without further fractionation by SEC. SAXS profiles were computed with the FoXS server^{1,2} for a series of models constructed from the continual spiroosome structure (two of which are shown in Fig. 4e). This analysis indicates that while the experimental data derive from a polydisperse AdhE sample, the overall shape of the SAXS curve is broadly consistent with that computed for a spiroosome model and is better described by the profile computed for the 24-mer model (the largest tested) than for the 12-mer or any lesser oligomers. Overall, these combined data show that AdhE forms into a spiroosome structure, which may be critical for its function.”

Fig 4e has been modified to include a 12-mer model and the computed curve for this 12-mer.

We cannot generate monodisperse spiroosome samples. We hope the revised text now conveys the absolute polydispersity of the sample from which the data in Fig. 4e arise. We are not suggesting that the 24-mer model is the model that arises from fitting these data - since this was not the analysis undertaken (i.e. we did not undertake *ab initio* modelling or even SASREF modelling based on the monomer). We were trying to demonstrate qualitatively that the data are consistent with spiroosome structures and *more* consistent with a lengthening spiroosome structure than a short one. We are cognisant of the disagreement between the SAXS curve computed for the 24-mer and the experimental data at very low q and agree that this indicates that there are significantly larger species in solution. Indeed, we have computed with SAXSMoW a MW equivalent to a 56 mer for the SAXS curve in Fig 4e. We could construct models larger than the 24-mer but, knowing that the data arise from a polydisperse sample, we do not want to overinterpret them.

-Fig 5; while I agree that there is a conformational change, I don't necessarily agree that the feature correlates directly to a real space change from ~80 angstrom to ~70 angstrom as asserted on page 9, line 175. SAXS profiles are the Fourier transform of all distances inside a particle, and features such as these can arise from form factors (spheres, cylinders etc...) in a way that is related to the overall size of particle. This is not diffraction data, where the peak position arises from d -spacing in electron density, and without a full analytical model of the data no conclusion can really be drawn about what real space physical distance(s) give rise to this feature. Further, the number of curves in panel b obscures the differences in the data, and really makes panel b not very informative nor supportive of the conclusions drawn by the authors.

→ We understand well that it is not possible to conclude which real space physical feature has given rise to the feature observed in the SAXS data. But we said that the feature **is consistent with** the helical pitch of the cryo-EM structure of spiroosomes, which it is. We refer this reviewer to the recent paper by Burian and Amenitsch³ in which the authors develop a dummy atom-based approach to the modelling of helical nanostructures and in which they frequently refer to the correlation between real-space features and the appearance of features in $p(r)$ functions (see e.g. Figure S12 of the paper by Burian and Amenitsch).

We could reduce the number of curves in panel b, but originally included the 4 cofactor curves to demonstrate that this change in feature is observed in all cofactor conditions. We feel it is supportive of the conclusions since, in all 4 cases, the feature is at a q different from that observed for the apo form. We have included SAXS data for spiroosome fraction 2 in new Supplementary Fig. 6 to show that the conformational change is also observed in the fraction comprising shorter spiroosomes.

-As an aside, the ApoAdhEk12 data (the orange curve) looks to my eye to be somewhat over subtracted. Buffer matching would be better supported if a concentration dependence for these data were provided in the supplemental data. It is conceivable that the large change in comparison the other curves could be caused by a poor background subtraction.

→ We have returned to the original SAXS data analysis and are confident that the ApoAdhEk12 data are not over-subtracted. A concentration series for the \pm cofactor data was not acquired. However, we were able to examine the integrated area plots generated as part of the volume of correlation analysis by ScÅtter (<http://www.bioisis.net>) which, for a correctly subtracted dataset, should flatten to a plateau at high q . We present here the integrated area plots for the data in Fig. 5b and for new Supplementary Fig. 6 (for spiroosome fraction 2, comprising shorter species) in which plateaus are clearly reached at high q . Thus, we are confident that the large change in comparison with the other curves is not caused by poor background subtraction, but is instead a real conformational change that is independent of spiroosome length (it being observed in both spiroosome fractions).

Integrated area plots for the data in Fig. 5b (left hand panel above) and for new Supplementary Fig. 6 (for spiroosome fraction 2, comprising shorter species, right hand panel above) in which plateaus are clearly reached at high q , indicating satisfactory buffer subtraction.

-Fig 6. The AUC data, in the form of the $c(s)$ distribution looks wrong. There's either too few points in the $c(s)$ distribution, or the regularization has not worked as there should be no sharp, jagged, peaks in a $c(s)$ distribution like the ones labeled 1 and 2. There's no primary data from the AUC runs included in the manuscript (such as the absorbance (or interference) v radial position over time plots) and no evidence of the quality of the $c(s)$ model fit. The authors don't quote a frictional ratio, which has a significant impact on the data. Was this parameter fixed for all samples? Was it allowed to float? How was the impact of large, very slowly diffusing particles on the smaller faster diffusing particles taken into account?

→ Peaks 1 and 2 in the original Fig. 6 were jaggy because, although a resolution of 200 was used, the data were analysed over a very wide s -range (0-120 S). We used this very wide range because the AdhE YP was unfractionated and included some very high- s species; we analysed the fractionated F670VAE mutants over the same s -range for consistency. We showed only a subset of the $c(s)$, for clarity, since the point of the figure is to illustrate the difference at low sedimentation coefficients between the mutant and WT samples. Peak 3 was as jaggy as 1 and 2, but the jaggiess was not apparent since the peak height was less.

Revisiting this figure in response to this reviewer's comments, we elected to re-analyse the raw data over a narrower s -range (0-20 S) and to exclude the first 5 scans from the WT data set (so that the very high- s species would have traversed the solution column to the cell base). The resultant $c(s)$ analysis is presented in revised Fig. 6d wherein a significant decrease in jaggiess is evident but there is no significant change in peak position.

We have included the primary data from the AUC runs as new Supplementary Fig. 8 which also includes the quality of the c(s) model fits.

We undertook c(s) analysis with the frictional ratio allowed to float and with it fixed (at a value of 1.3, wherein the baseline, meniscus and cell base positions were floated). Whether it was fixed or floated actually makes little difference to the quality of the fits or to the results obtained - the peaks in the c(s) are perfectly superimposable.

The reviewer makes an excellent point regarding the impact of large, very slowly diffusing particles on the smaller faster diffusing particles. We had not explicitly considered this in the original manuscript. On reflection, we realise that the sedimentation coefficients of the small species in the 3 mutant samples (F670VAE) will have been less affected by their passage through a background of larger species than is the case for the AdhE YP, since this latter sample was not pre-fractionated in any way and so contained a greater number of large species. Thus, we expect the sedimentation coefficients of the smaller species in AdhE YP to be smaller than they should be. This means that it is possible that the sedimentation coefficient of peak 3 in Fig 6d is slightly *underestimated*. **This strengthens our assertion that monomer species are not observed in solution in the absence of the F670VAE mutation.**

- 1 Schneidman-Duhovny, D., Hammel, M., Tainer, John A. & Sali, A. Accurate SAXS profile computation and its assessment by contrast variation experiments. *Biophys. J.* **105**, 962-974, doi:<https://doi.org/10.1016/j.bpj.2013.07.020> (2013).
- 2 Schneidman-Duhovny, D., Hammel, M., Tainer, J. A. & Sali, A. FoXS, FoXSDock and MultiFoXS: Single-state and multi-state structural modeling of proteins and their complexes based on SAXS profiles. *Nucleic Acids Res.* **44**, W424-W429, doi:10.1093/nar/gkw389 (2016).
- 3 Burian, M. & Amenitsch, H. Dummy-atom modelling of stacked and helical nanostructures from solution scattering data. *Iucrj* **5**, 390-401, doi:10.1107/s2052252518005493 (2018).

REVIEWERS' COMMENTS:

Reviewer #2 (Remarks to the Author):

My comments were only minor and have been addressed appropriately,

Reviewer #3 (Remarks to the Author):

The authors have made a significant effort to include most of the changes I suggested, and have mostly addressed my concerns. I approve of the change in wording for the text related to figure 4, which certainly makes it clear that the result is affected by polydispersity; and agree that the AUC data included is now sufficient to assess the conclusions.

I'm still concerned about the background subtraction, but the authors have clearly made an effort to not over interpret this data; although I would still be hesitant to conclude a real space distance directly from such a feature in a SAXS curve, and would generate a $P(r)$, which definitely correlates strongly to real space distances, to support the conclusion. However given the nature of the sample this may not be possible, and the results are mostly consistent with a shape change, so I will accept the conclusion as it stands.